# Homologous Recombination Deficiency and Cyclin E1 Amplification Are Correlated with Immune Cell Infiltration and Survival in High-Grade Serous Ovarian Cancer

**DOI:** 10.3390/cancers14235965

**Published:** 2022-12-02

**Authors:** Lilian van Wagensveld, Juliette O. A. M. van Baal, Maite Timmermans, Duco Gaillard, Lauri Borghuis, Seth B. Coffelt, Efraim H. Rosenberg, Christianne A. R. Lok, Hans W. Nijman, Loes F. S. Kooreman, Joyce Sanders, Marco de Bruijn, Lodewyk F. A. Wessels, Rianne van der Wiel, Christian Rausch, Annegien Broeks, Roy F. P. M. Kruitwagen, Maaike A. van der Aa, Gabe S. Sonke, Philip C. Schouten, Koen K. Van de Vijver, Hugo M. Horlings

**Affiliations:** 1Department of Research and Development, Netherlands Comprehensive Cancer Organization (IKNL), 3511 DT Utrecht, The Netherlands; 2Department of Molecular Pathology, The Netherlands Cancer Institute, 1066 CX Amsterdam, The Netherlands; 3GROW, School for Oncology and Reproduction, 6229 HX Maastricht, The Netherlands; 4Department of Gynecology, Center for Gynecologic Oncology Amsterdam (CGOA), 1066 CX Amsterdam, The Netherlands; 5Department of Obstetrics and Gynecology, Leiden University Medical Centre, 2333 ZA Leiden, The Netherlands; 6Department of Molecular Carcinogenesis, The Netherlands Cancer Institute, 1066 CX Amsterdam, The Netherlands; 7Division of Tumor Biology & Immunology, The Netherlands Cancer Institute, 1066 CX Amsterdam, The Netherlands; 8Institute of Cancer Sciences, University of Glasgow, Glasgow G12 8QQ, UK; 9Cancer Research UK, Beatson Institute, Glasgow G61 1BD, UK; 10Department of Pathology, The Netherlands Cancer Institute, 1066 CX Amsterdam, The Netherlands; 11Department of Obstetrics and Gynecology, University Medical Center Groningen, University of Groningen, 9700 RB Groningen, The Netherlands; 12Department of Pathology, Maastricht University Medical Centre, 6229 HX Maastricht, The Netherlands; 13Department of Pathology, VU University Medical Center, 1081 HV Amsterdam, The Netherlands; 14BioLizard nv, 9000 Ghent, Belgium; 15Core Facility Molecular Pathology & Biobanking, The Netherlands Cancer Institute, 1066 CX Amsterdam, The Netherlands; 16Department of Obstetrics and Gynecology, Maastricht University Medical Centre, 6229 HX Maastricht, The Netherlands; 17Department of Medical Oncology, The Netherlands Cancer Institute, 1066 CX Amsterdam, The Netherlands; 18Department of Pathology & Cancer Research Institute Ghent (CRIG), Ghent University Hospital, 9000 Ghent, Belgium

**Keywords:** epithelial ovarian carcinoma, microenvironment, tumor, recombination, homologous, prognosis, ovarian neoplasms/genetics

## Abstract

**Simple Summary:**

Ovarian cancer is the deadliest gynecological cancer in developed countries of which high-grade serous ovarian carcinoma (HGSOC) is the most common subtype. How the tumor’s genetic characteristics are associated with the tissue surrounding the tumor; the tumor microenvironment (TME), is incompletely understood. Our study assessed the TME and genetic profiles of HGSOC and their associations with survival. 347 patients with HGSOC were categorized in the following profiles: *BRCA* mutation (*BRCA*m) (30%), non-*BRCA* mutated homologous recombination deficiency(HRD) (19%), *CCNE1*-amplification (13%), non-*BRCA*mut HRD and *CCNE1*-amplification (double classifier) (20%), and no specific molecular profile (NSMP) (18%). *BRCA*m profile showed the best survival and *CCNE1* and double classifier the worst. Higher immune cell densities showed a favorable survival, also within the molecular profiles. Furthermore, immune cell densities differed per molecular profile with *BRCA*m profile tumors showing the highest and *CCNE1* lowest densities. Our study showed that HGSOC is not one group but is grouped by different molecular profiles and TME.

**Abstract:**

Background: How molecular profiles are associated with tumor microenvironment (TME) in high-grade serous ovarian cancer (HGSOC) is incompletely understood. Therefore, we analyzed the TME and molecular profiles of HGSOC and assessed their associations with overall survival (OS). Methods: Patients with advanced-stage HGSOC treated in three Dutch hospitals between 2008–2015 were included. Patient data were collected from medical records. *BRCA1/2* mutation, *BRCA1* promotor methylation analyses, and copy number variations were used to define molecular profiles. Immune cells were assessed with immunohistochemical staining. Results: 348 patients were categorized as *BRCA* mutation (*BRCA*m) (*BRCA*m or promotor methylation) (30%), non-*BRCA* mutated HRD (19%), Cyclin E1 (*CCNE1*)-amplification (13%), non-*BRCA*mut HRD and *CCNE1*-amplification (double classifier) (20%), and no specific molecular profile (NSMP) (18%). *BRCA*m showed highest immune cell densities and *CCNE1*-amplification lowest. *BRCA*m showed the most favorable OS (52.5 months), compared to non-*BRCA*mut HRD (41.0 months), *CCNE1*-amplification (28.0 months), double classifier (27.8 months), and NSMP (35.4 months). Higher immune cell densities showed a favorable OS compared to lower, also within the profiles. CD8+, CD20+, and CD103+ cells remained associated with OS in multivariable analysis. Conclusions: Molecular profiles and TME are associated with OS. TME differs per profile, with higher immune cell densities showing a favorable OS, even within the profiles. HGSOC does not reflect one entity but comprises different entities based on molecular profiles and TME.

## 1. Introduction

Worldwide approximately 295,000 women are diagnosed with ovarian cancer annually, and 185,000 women die due to the disease, making ovarian cancer the most deadly gynecologic malignancy in developed countries [1]. High-grade serous ovarian carcinoma (HGSOC) is the most common subtype of epithelial ovarian cancer (EOC) (approximately 70% of all EOCs) and accounts for 70–80% of all ovarian cancer deaths [2]. Even though HGSOC initially shows good response rates to platinum- and taxane-based chemotherapy, disease recurrence is frequent and often chemotherapy-resistant [2]. 

Large-scale genomic and epigenomic studies revealed that HGSOC is characterized by extensive copy number variations (CNV), high genomic instability, and clonal diversity [3,4,5]. The genetic makeup of HGSOC is associated with both distinct clinical and biological characteristics. 

Mutational and functional alterations in genes that are involved in homologous recombination repair (HRR) mechanisms are found in approximately 50% of HGSOC [2,3,6]. The majority of these homologous repair deficient (HRD) tumors exhibit *BRCA1* and *BRCA2*-gene deficiencies [3]. *BRCA1* and *BRCA2* mutations are associated with higher response rates to platinum-based chemotherapy and PARP inhibitors (PARPi), and with longer survival compared to their wild-type counterparts [7,8,9]. Platinum-based therapy and PARPi exploit HRD, platinum-based therapy by inducing double-strand breaks in DNA, and PARPi by impeding tumor DNA repair via synthetic lethality. It is hypothesized that HRD tumors exhibit a high mutational load resulting in higher levels of neo-antigens. This, in turn, increases tumor-cell recognition by T-cells, facilitating an effective lymphoid immune response and also resulting in favorable survival [9]. 

HGSOC can also exhibit a histological and prognostic phenotype similar to the phenotype seen in *BRCA1/2* mutation carriers called “*BRCA*-ness” [10]. *BRCA*-ness refers to the phenotypic characteristics of tumors lacking *BRCA1/2* germline mutations that exhibit defects in HRR mimicking *BRCA* loss, per instance due to alterations in RAD51 or epigenetic silencing of *BRCA1* [3,10,11]. Another way through which tumors can resemble *BRCA*-mutated tumors, is by epigenetic inactivation of the *BRCA* gene without alterations to its DNA sequence. Normally, the regulatory region of the full active *BRCA1* gene is de-methylated. Methylation of the *BRCA1* promotor leads to the incapability of *BRCA1* gene transcription and therefore inactivation of the gene [12]. Hypermethylation of the *BRCA1* gene-promotor occurs in 10 to 20% of EOCs [13], and such patients show a superior survival compared to patients with an unmethylated BRCA1 gene-promotor [14]. Hypothetically, *BRCA*-methylated EOCs could be a new subset of cancers with impaired *BRCA* function [14]. However, the 2020 ESMO recommendation stated that not enough evidence is available to determine the clinical validity of *BRCA1* promoter methylation yet [15].

Another important genetic subgroup of HGSOC is represented by focal gene amplification of *Cyclin E1* (*CCNE1*), present in approximately 20% of HGSOC [3]. *CCNE1* is involved in tumorigenesis via induction of chromosomal and genetic instability [16]. Remarkably, previous studies demonstrated that *CCNE1* amplification and *BRCA* mutations are largely mutually exclusive [3,6]. HGSOC with *CCNE1* amplification is correlated with poor survival and primary resistance to platinum-based chemotherapy [6,17,18].

How the genetic makeup of HGSOC influences immune cell infiltration is not completely understood. To establish new prognostic biomarkers for patient outcome, as well as identify potential future therapeutic targets for different subtypes of HGSOC, knowledge of the underlying influences forming these subtypes is a prerequisite. Therefore, our study aims to assess the association of HRD and *CCNE1* amplification in HGSOC with infiltration of T-cells (CD8 and CD103), B-cells (CD20), and macrophages (CD68) that have previously been associated with survival in HGSOC [19,20,21,22,23]. In addition, we analyzed whether molecular profile and immune cell infiltration are independently correlated with survival.

## 2. Materials and Methods

### 2.1. Patient and Tumor Selection

Patients with HGSOC International Federation of Gynecology and Obstetrics (FIGO) stage IIb-IV, who were treated with primary cytoreductive surgery (PDS) and adjuvant chemotherapy in one of three Dutch tertiary referral hospitals (Netherlands Cancer Institute—Antoni van Leeuwenhoek Hospital (NKI-AVL), Maastricht University Medical Centre (MUMC) and Amsterdam University Medical Centre (AUMC)), between January 2008 and December 2015, were eligible for the present study. Furthermore, patients from NKI-AVL treated with neoadjuvant chemotherapy (NACT) followed by interval debulking surgery (IDS) and adjuvant chemotherapy, were also included. Patients were excluded in case no tumor tissue was available for immune cell analyses or molecular analyses.

Clinical data were extracted from the Netherlands Cancer Registry (NCR) and histopathological data from the Dutch Pathology Registry (PALGA). The NCR is a nationwide registry managed by the Netherlands Comprehensive Cancer Organization (IKNL) and covers all primary malignancies in the Netherlands since 1989. The following parameters were extracted from the NCR; performance status, germline *BRCA* status, treatment sequence (NACT, NACT-IDS, or PDS), surgery outcome (complete with no visible disease; optimal with ≤1 cm residue or sub-optimal with >1 cm residue), and data on progression. Progression of disease was defined in case of symptoms combined with increased serum CA-125 levels, radiological signs of progression, or histological or cytological confirmation of recurrent disease. Vital status and date of death were obtained by the NCR via linkage with the municipal population registration. Pathological data and tumor tissue blocks were obtained from the nationwide network PALGA, which registers all records of histopathology and cytopathology with full coverage since 1991 [24].

### 2.2. Tissue Samples and Tissue Microarrays (TMA)

Formalin-fixed, paraffin-embedded (FFPE) tissue blocks from all patients with primary HGSOC were obtained. Tissue blocks originated from samples retrieved during debulking surgery, which subsequently resulted in pretreated tumor blocks in case a patient received NACT. All cases underwent pathological re-review based on conventional morphological examination of sections stained with hematoxylin and eosin (H&E) by three dedicated pathologists (K.V.d.V., H.H., J.S.). Tumor grade was designated according to the binary grading classification (to exclude low-grade serous ovarian carcinoma) [25].

The paraffin tissue blocks were organized into TMAs. Representative areas of the center and peripheral invasive margin of the ovarian tumor were selected on whole-tissue FFPE H&E stained slides for immune and tumor cell scoring. In case ovarian tumor tissue was not available a representative tumor block from another location (peritoneum, omentum) was selected. In each tumor four cores were selected, optimally representing tumor and peripheral stroma containing immune cell infiltrate. TMAs with one mm-sized cores were constructed using a tissue microarrayer (Grand Master, Sysmex Europe GmbH, Norderstedt, Germany). To enable adhesion of the cores to the recipient paraffin block, the block was melted at 70 °C for nine minutes and cooled down overnight.

### 2.3. Immunohistochemical (IHC) Staining

CD8+, CD20+, CD68+, and CD103+ cell expression was assessed by IHC (Figure 1). IHC was performed on the BenchMark Ultra autostainer (Ventana Medical Systems Inc., Oro Valley, AZ, USA). Three μm thick TMA sections were generated and heated at 75 °C for 28 min followed by deparaffinization and rehydration. Deparaffinization was completed in the instrument using an EZ prep solution (Ventana Medical Systems Inc., Oro Valley, AZ, USA). Heat-induced antigen retrieval was initiated using Cell Conditioning 1 (Ventana Medical Systems Inc., Oro Valley, AZ, USA) for 32 min at 95 °C. CD8+ was detected using clone C8/144B (1/200 dilution, M7103, Agilent Technologies, Santa Clara, CA, USA), CD20+ was detected using clone L26 (1/800 dilution, M0755, Agilent Technologies, Santa Clara, CA, USA), CD68+ was detected using clone KP1 (1/20000 dilution, M0814, Agilent Technologies, Santa Clara, CA, USA). CD103+ staining was performed using anti-αEβ7-integrin (1/200 dilution, ab129202, Abcam, Cambridge, UK), as described by Komdeur et al. [26]. The bound antibodies were detected using the OptiView DAB Detection (Ventana Medical Systems Inc., Oro Valley, AZ, USA). Slides were counterstained with Hematoxylin II and Bluing reagent (Ventana Medical Systems Inc., Oro Valley, AZ, USA). All stained TMA slides were digitalized with a 20× magnification, using Leica Aperio AT2 Digital Pathology Slide Scanner (Leica Microsystems, Wetzlar, Germany). Total numbers of CD8+, CD20+, CD68+, and CD103+ cells were manually counted per core and scored as 0, 1–5, 6–19, 20–49, 50–100, or >100 positive cells. The highest count per tumor was used. Immune cells were further categorized into low, medium, and high densities based on the 25th and 75th percentiles and the median. For CD103+ cells, only intraepithelial-located cells were counted and analyzed. All slides were counted manually by 2 individuals, differences in counts of over 10% were reanalyzed and discussed until a consensus was reached. Cores without tumor tissue were excluded from the final analyses.

### 2.4. Molecular Analyses

DNA isolation from FFPE tissue blocks with a minimal tumor percentage of 20% was performed fully automated according to standard protocols using the Qiacube (Qiagen, Hilden, Germany). Ten serial sections of 10 µm thickness of each tumor were taken using a Finesse ME+ microtome (Thermo Fisher Scientific, Waltham, MA, USA) and deparaffinized using the ST5020 multi-stainer (Leica Microsystems, Wetzlar, Germany). DNA isolation was performed with AllPrep RNA/DNA FFPE Kit (Qiagen, Hilden, Germany). The manufacturer’s instructions were followed for DNA isolation using the Qiacube (Qiagen, Hilden, Germany). Quantification of the concentration and purity of all DNA extracts was performed using a NanoDrop-8000 spectrophotometer (Thermo Fisher Scientific, Waltham, MA, USA). Measurement of double-stranded DNA (dsDNA) yield was performed with a Qubit dsDNA High Sensitivity Assay Kit (Thermo Fisher Scientific, Waltham, MA, USA).

Molecular profiles were determined in a stepwise manner. Germline *BRCA* mutation information was obtained from the NCR database. In all patients without a known germline *BRCA* mutation, tumor *BRCA* mutation was determined. In cases without *BRCA* mutation, *BRCA1* promotor methylation status was determined. Patients without a *BRCA* mutation or *BRCA1* promotor methylation were further analyzed with CNV sequencing for non-*BRCA* mutation (non-*BRCA*mut) HRD profile and *CCNE1* amplification based on low-coverage whole-genome sequencing data. This stepwise manner was used as *BRCA1* methylation and *CCNE1* amplification are mutually exclusive with *BRCA* mutation and as *BRCA*-ness refers to non-*BRCA* mutated tumors mimicking *BRCA* loss [3,10,13]. In the remaining patients in which none of the aforementioned molecular profiles were found were categorized as “no specific molecular profile” (NSMP).

### 2.5. Somatic BRCA Analysis

Quality control of DNA was performed using the QC plex kit (Agilent Technologies, Santa Clara, CA, USA). The manufacturer’s instructions were followed, using the 2100 bioanalyzer (Agilent Technologies, Santa Clara, CA, USA). *BRCA1* and *BRCA2* somatic mutation analyses were performed with the BRCA MASTR Plus Dx kit (Agilent technologies, Santa Clara, CA, USA) according to the manufacturer’s instructions, using a Veriti Thermocycler (Thermo Fisher Scientific, Waltham, MA, USA) and the MiSeq (Illumina, San Diego, CA, USA). The results of the MiSeq run were uploaded to the MASTR reporter (Agilent technologies, Santa Clara, CA, USA) for further analysis.

### 2.6. BRCA Promotor Methylation

To determine *BRCA1* promotor methylation status, Methylation-specific MLPA (MS-MLPA) was performed using a commercial kit (ME0053 kit, MRC Holland, Amsterdam, The Netherlands). The manufacturer’s instructions were followed to determine hypermethylation with the use of the Veriti thermocycler (Thermo Fisher Scientific, Waltham, MA, USA). Fragment analysis was performed using the genetic analyzer (ABI-3500, Thermo Fisher Scientific, Waltham, MA, USA). The probes used were designed to contain a HhaI recognition site (GCGC) and thus target one CpG dinucleotide within a CpG island. If the HhaI recognition site is not methylated, HhaI will cut the probe-sample DNA hybrid and no PCR product will be formed. If the target DNA is methylated the fragment will be amplified during subsequent PCR [27].

### 2.7. Low-Coverage Whole-Genome Sequencing (WGS)

The total amount of DNA was quantified on the Nanodrop 2000 (Thermo Fisher Scientific, Waltham, MA, USA), and the amount of double-stranded DNA in the genomic DNA samples was quantified with the Qubit dsDNA HS Assay Kit (Invitrogen, Waltham, MA, USA, cat no Q32851). Because of variable DNA extraction efficiencies or sample sizes, varying input amounts of double-stranded DNA were used (from 34 ng to 964 ng). These quantities were Covaris sheared in a standard volume of 130 μL, bead cleaned, and eluted in 50 ul elution buffer that was used to the full extent to start library preparation. Samples were purified using 2× Agencourt AMPure XP PCR Purification beads according to the manufacturer’s instructions (Beckman Coulter, Brea, CA, USA, cat no A63881). Sheared samples were quantified and qualified on a BioAnalyzer system using the DNA7500 assay kit (cat no. 5067-1506, Agilent Technologies, Santa Clara, CA, USA). Library preparation for Illumina sequencing was performed with a maximum input of 1 μg sheared DNA using the KAPA Hyper Prep Kit (KK8504, KAPA Biosystems, Wilmington, MA, USA). During library enrichment, 6 PCR cycles were used to obtain enough yield for sequencing. After library preparation, the libraries were cleaned up using 1× AMPure XP beads. Libraries were analyzed with a BioAnalyzer system using DNA7500 chips to define molarity. Three pools were created, two with ninety-three and one with sixty-three uniquely indexed samples were mixed together by equimolar pooling, in a final concentration of 10 nM, and subjected to sequencing on an Illumina HiSeq2500 machine in a total of twenty-two lanes of a single read 65 bp run, according to the manufacturer’s instructions.

### 2.8. Non-BRCAmut HRD Classification

A non-*BRCA*mut HRD copy number profile was determined according to methods described previously by Schouten et al. [28]. In short, reads were aligned to the reference genome GRCh38 using BWA-MEM (version 0.7.17). Reads with a mapping quality of over 15 were counted in 20 kb non-overlapping bins, corrected for CG bias, and corrected for local alignment-bases estimated mappability, resulting in ^2^log count ratios. The 20 kb resolution ^2^log ratios were mapped to the 1 MB resolution input for the classifier. Subsequently, we corrected the centering and scaling of the data between the sequencing platform of the current study and the oligonucleotide array platform on which the classifier was created. We fitted a linear regression model with Gaussian distribution and identity link function using the R glm function to the sorted location-wise average of the training set and the current dataset. The obtained alpha coefficient to correct the centering and the obtained beta coefficient to correct the scaling of the current data.

All genomic profiles underwent automated and manual quality control. The profiles were subsequently classified using the described shrunken centroids classifier. Profiles with a posterior probability of >0.5 were classified as non-*BRCA*mut HRD, and profiles with a posterior probability of ≤0.5 were classified as non-*BRCA*-like.

### 2.9. CCNE1 Classification

A FastQC [29] report was generated for quality control of sequencing reads and further checked in MultiQC [30]. Reads with adapter sequences were trimmed using Trimmomatic [31], and correct trimming was confirmed with a second MultiQC report. Reads were aligned to the hg19 reference genome using BWA mem, and subsequently sorted and indexed using Samtools [32]. Duplicates were marked using Picard MarkDuplicates [33]. Copy Number Profiles were generated using the QDNAseq suite [34]. Reads were binned into 30 kb-sized bins. Blacklisted bins were removed using default filtering. Bins were corrected for GC-content and mappability. Finally, read counts were log2 normalized and outliers were removed. Standard QDNAseq segmentation quality using Circular Binary Segmentation (CBS) resulted in noisy segmentation in some cases. Noisy samples were smoothed (Appendix A) while retaining unbiased segmentation in high-quality profiles. Finally, Copy numbers were called using QDNAseq, using its implementation of CGHcall.

### 2.10. Statistical Analyses

Statistical analyses were performed in STATA/SE (version 14.1, STATA CORP, College Station, TX, USA). A *p*-value < 0.05 was considered significant. The influence of molecular profile was explored by the unique profiles: *BRCA* mutation (*BRCA*m) profile (*BRCA* 1/2 mutation or promotor methylation), non-*BRCA*mut HRD, *CCNE1* gain/amplification, double classifier (non-*BRCA*mut HRD and *CCNE1* gain/amplification), and NSMP. Basic patient characteristics and immune cell densities of these profiles were assessed with Chi-square tests for categorical variables, One-way ANOVA for normally distributed continuous variables, and Kruskal–Wallis for non-normally distributed continuous variables. Ordinal logistic regression was used to investigate the influence of molecular profiles on immune cell densities and the Chi-square test to investigate the relationship between tumor regression and immune cell density groups. Kaplan–Meier survival estimates with the corresponding logrank test and univariable and multivariable Cox regression analyses were used to assess the effect of molecular profile on progression-free survival (PFS) and overall survival (OS). Those found significant in univariate analyses with a *p* < 0.10, were included in the multivariable regression analyses and assessed using backward selection. PFS was calculated as the time between the start of primary treatment and the date of recurrence or date of death of disease (DoD). OS was calculated as the interval between the start of treatment and the DoD, or if alive, the date of the last check of the municipal population register (31 January 2021). In case no event had occurred (recurrence nor death), patients were right censored at the time of the last follow-up.

## 3. Results

### 3.1. Patient Characteristics

We selected patients with advanced-stage HGSOC treated in one of three Dutch tertiary referral hospitals (NKI-AVL, MUMC, and AUMC) between 2008–2015. Our final cohort consisted of 360 patients with advanced-stage HGSOC (Appendix A, Flowchart). Patients were categorized into five molecular profiles, consisting of (1) *BRCA*m profile (tumor *BRCA*m or *BRCA1* promotor methylation), (2) non-*BRCA*mut HRD, (3) *CCNE1* gain/amplification, (4) double classifier (non-*BRCA*mut HRD and *CCNE1* gain/amplification), and (5) NSMP. Table 1 demonstrates the clinical characteristics of the patient cohort stratified by molecular profiles.

### 3.2. Molecular Profiles

We were able to assess the molecular profile in 348 out of 360 patients. Twelve patients were excluded as a result of insufficient DNA quality. The molecular profiles are depicted in Figure 2. In 78 out of the 348 patients (22%), a *BRCA* mutation was present: 25 and 25 patients with a germline or somatic *BRCA1* mutation, respectively, and 18 and 10 patients with a germline or somatic *BRCA2* mutation. 27 patients (8%) had a *BRCA1* promotor methylation resulting in a total 105 patients (30%) with a *BRCA*m profile. In 67 patients (19%) a non-*BRCA*mut HRD profile was found. In 45 patients (13%) increased *CCNE1* copy numbers (gain *n* = 28; amplification *n* = 17) were detected. 69 patients (20%) depicted both a non-*BRCA*mut HRD profile and increased *CCNE1* copy numbers. 62 patients (18%) had no specific molecular profile.

### 3.3. Immune Cell Infiltration in Molecular Profiles

Table 2 lists the immune cell densities, stratified by molecular profiles. The number of immune cells did not significantly differ between the molecular profiles, except for CD68+ cells, which were highest in *BRCA*m profile tumors (>100 cells 44.8%, compared to 16–22% in other molecular profiles). Although non-significant, compared to the other profiles *BRCA*m profile tumors showed higher amounts of CD8+ cells (32.4% > 100 cells), CD103+ cells (27.6% > 100 cells) and CD20+ cells (20.9% > 50 cells). Tumors with a *CCNE1* amplification/gain showed the lowest amount of these immune cells (>100 CD8+ cells 20.0%; >100 CD103+ cells 11.1%; >50% CD20+ cells 4.4%).

### 3.4. PFS and OS of Molecular Profiles and Immune Cell Infiltration

The median follow-up of the total cohort was 38.2 months (IQR 22–66). Kaplan–Meier curves showed a significant association between molecular profile and overall survival (OS) (Logrank = 0.0003) (Figure 3A). Median OS was most favorable in patients with a *BRCA*m profile (52.5 months), followed by non-*BRCA*mut HRD (41.0 months; HR 1.31; 95%CI 0.93–1.85, compared to *BRCA*m profile) and NSMP (35.4 months; HR 1.67; 95%CI 1.18–2.38) (Figure 3B). OS was similar for the patients with *CCNE1* gain/amplification or double classifier (28.0 months; HR 2.17; 95%CI 1.48–3.18 and 27.8 months; HR 1.75; 95%CI 1.24–2.46, respectively). Kaplan–Meier curves also showed a significant association between molecular profile and PFS (Logrank = 0.0013). Median progression free survival (PFS) was 22.3 months in patients with a *BRC*Am profile, 16.9 months in non-*BRC*Amut HRD (HR 1.28; 95%CI 0.89–1.86, compared to *BRCA*m profile), 14.8 months in *CCNE1* gain/amplification (HR 1.94; 95%CI 1.29–2.91), 14.7 months in double classifier (HR 1.90; 95%CI 1.33–2.72), and 16.7 in NSMP (HR 1.60; 95%CI 1.12–2.30) (Table 3).

Kaplan–Meier curves were generated for OS per immune cell type (Figure 4) and further subgrouped by the two treatment groups: PDS and NACT (Appendix A). Defined as high and low densities, CD8+, CD68+, and CD103+ cells showed a significant association with survival (Logrank; *p* = 0.0171, *p* = 0.0151, *p* = 0.0015, respectively). In all cases, a lower density resulted in poorer survival. Lower densities of CD20+ cells also showed a poorer survival, yet non-significant. Median OS was most favorable in patients with higher immune cell densities (CD8+ cells; 44.2 vs. 36.3 months; HR 0.73; 95%CI 0.56–0.95, CD20+ cells; 44.2 vs. 36.9 months; HR 0.72; 95%CI 0.52–1.00, CD68+ cells; 47.7 vs. 36.8 months; HR 0.72; 95%CI 0.56–0.94 and CD103+ cells; 52.1 vs. 36.3 months; HR 0.61; 95%CI 0.45–0.83, respectively) (Table 4). Within the treatment types a significant association between immune cell densities and OS was seen in case of PDS in CD8+, CD20+, and CD103+ cells (Logrank; *p* = 0.0093, *p* = 0.0148, *p* = 0.0180, respectively). In NACT there was also a favorable survival in case of higher immune cell densities, yet non-significant (Appendix A). Furthermore, an association was also seen within the different molecular profiles. Although predominantly non-significant, presumably due to low numbers, higher immune cell densities showed the tendency of a favorable OS within the several molecular profiles, except for CD20+ cells in *BRCA*m profile patients (Appendix A). Higher immune cell densities also resulted in a favorable PFS (Appendix A).

### 3.5. Multivariable Analyses

After adjustment for age, FIGO stage, therapy sequence, and completeness of debulking surgery in multivariable cox regression, the associations of molecular profile with OS remained consistent, except for NSMP (Figure 3B). Compared to the *BRCA*m profile group, *CCNE1* amplification and double classifier profile correlated significantly with poorer OS (HR 1.75; 95%CI 1.16–2.65 and HR 1.53; 95%CI 1.07–2.18, respectively). The same accounted for PFS (Table 3). Compared to the *BRCA*m profile group, *CCNE1* amplification and double classifier profile both correlated significantly with poorer PFS (HR 1.57; 95%CI 1.02–2.43 and HR 1.77; 95%CI 1.22–2.56, respectively). Neither the immune cell densities nor the interaction between immune cell densities and molecular profiles were significant confounders.

Multivariable Cox regression showed that immune cell densities remained associated with OS after adjustment for age, FIGO stage, therapy sequence, and completeness of debulking surgery (Table 4). Higher CD8+, CD20+ and CD103+ densities resulted in a significant favorable OS compared to lower densities (HR 0.72; 95%CI 0.55–0.94, HR 0.71; 95%CI 0.51–0.98 and HR 0.68; 95%CI 0.50–0.93, respectively). CD68+ cells showed a favorable OS for higher densities in the univariate analysis but were non-significant in the multivariate analysis. Notably, in multivariable cox regression, only CD20+ densities remained associated with PFS after adjustment for age, FIGO stage, therapy sequence, and completeness of debulking surgery (Appendix A).

## 4. Discussion

Our study reports an improved OS in patients with a *BRCA* mutation or promotor methylation and a worse OS in patients with a *CCNE1* amplification/gain. This is in line with previous reports describing a correlation between OS and response to platinum-based chemotherapy, which is enhanced in *BRCA*-mutated tumors and decreased in *CCNE1* amplified tumors [17,18,35,36,37]. Furthermore our study depicted an improved OS in patients non-*BRCA*mut HRD supporting the hypothesis that non-*BRCA*mut HRD and possibly *BRCA1* promotor methylated EOCs are an important subset of cancers with impaired HRR [10,14].

Our study confirmed that immune cell infiltrates are associated with OS. Higher CD8+, CD20+, and CD103+ cell densities resulted in a more favorable survival compared to lower cell densities. These results are in agreement with previous studies [19,20,21]. This was also seen within the two treatment types, PDS and NACT, even though nonsignificant in case of NACT. The link between tumor infiltration with macrophages and patient survival is more complex with conflicting study results. Tumor-associated macrophages have been described to promote cancer progression [38,39]. In contrast, our cohort and other previous studies demonstrated an association between higher levels of macrophages and improved OS [22] while others show a negative or no influence on OS [23,40] insinuating a complex role of immune cells and other factors influencing OS.

We found that immune cells were most prominent in *BRCA* mutated or *BRCA1* promotor methylation patients. *BRCA* gene mutations have been correlated with increased immune cell infiltration in HGSOC [9,20,41]. Immune cell infiltration, in turn, has been associated with increased response to immunotherapy [42]. Although immunotherapy is not proven to be beneficial (yet) in the treatment of HGSOC, our results suggest that *BRCA* mutated HGSOC will be the most eligible candidates for immunotherapy in the future, in contrast to *CCNE1* amplified HGSOC.

In EOC patients, non-*BRCA*mut HRD has been correlated with a favorable response to platinum-based chemotherapy and PARPi [43], similar to the benefit that has been seen from DNA double-strand-break-inducing chemotherapy in breast cancer patients with non-*BRCA*mut HRD profiles compared to patients without this profile [44,45,46]. In Zhang et al. investigated non-*BRCA*mut HRD using whole-exome deep sequencing data from the TCGA and showed a similar OS in *BRCA* mutated and non-*BRCA*mut HRD ovarian cancer patients, which was significantly better compared to patients without HRD [47]. In the present study, we showed a moderate beneficial effect of non-*BRCA*mut HRD tumors on OS of approximately 6 months, whereas patients with a *BRCA*m profile showed a significant increase of 17 months in OS, compared to our NSMP group. The nonsignificance of the effect in patients with non-*BRCA*mut could be explained by the relatively good performance of the NSMP group, from which patients with a *CCNE1* gain/amplification, with a significantly worse OS, were excluded. Presumably, this explains the differences between our study and the results of non-*BRCA*mut HRD reported in the literature. Furthermore, the survival curve of the double classifier group tends to follow the survival curve of the *CCNE1* gain/amplification group and shows a similar median survival rate. This suggests a more dominant influence of *CCNE1* gain/amplification compared to the influence of non-*BRCA*mut HRD.

Remarkably, the immunological response in non-*BRCA*mut HRD tumors shares similarities with the response in tumors without a *BRCA* mutation. Hypothetically, non-*BRCA*mut HRD tumors display lower levels of neo-antigens, which leads to less tumor-cell recognition by T-lymphocytes, compared to *BRCA* mutated tumors. More clinical trials must confirm to what extent patients with non-*BRCA*mut HRD tumors are comparable to patients with tumors harboring somatic or germline *BRCA* mutations in clinical outcome and response to immunotherapy.

Activation of the RB1/*CCNE1* pathway is considered to be largely exclusive to *BRCA* mutations [3,48]. *CCNE1* activates transcription of *BRCA1* and *BRCA2* genes, thereby stimulating the HRR pathway, which gives *CCNE1* amplified tumors the ability to better withstand DNA double-strand-break systemic therapy [49]. Our results also suggest that *CCNE1* amplified tumors are characterized by relative chemotherapy resistance and therefore might require a different treatment approach. Moreover, we showed that immune cells are less abundant in this molecular profile, suggesting that patients with *CCNE1*-amplified tumors could be less likely to respond to immunotherapy. Other treatment strategies for *CCNE1* amplified, *BRCA*-wildtype tumors have not been investigated in clinical trials yet. However, in both in vitro and in vivo studies, *CCNE1* amplified HGSOC showed sensitivity to Cyclin-dependent kinase (CDK)-inhibitor dinaciclib and showed synergism with AKT-inhibitor MK-2206 [50].

Even though immune cell densities differed between the distinct molecular profiles, possibly warranting different treatment approaches, the tendency of a favorable OS for patients with higher immune cell densities was seen within molecular profiles as well. This finding indicates that even within the favorable, or non-favorable, molecular profiles immune cell densities strongly influence survival.

The strength of the present study is that we integrated detailed genomic, immunological, and clinical data from a large cohort of patients with HGSOC. Our data defines distinct prognostic profiles of HGSOC based on molecular and immune profiles. A central pathology rereview of all histological tissue was performed by dedicated pathologists in gynecologic oncology. Lastly, a long follow-up was achieved, and clinical data were complete. This study is not without limitations. A limitation is that immune cells were scored on tissue derived from PDS, but also after NACT. Pre-NACT immune cell levels have shown to be different from post-NACT levels, predominantly with higher levels post-NACT [51,52]. Ideally, immune cells are scored before, during, and after treatment with NACT. Nonetheless, pre- and post-NACT immune cells have been associated with OS and PFS in the same manner, with higher levels being associated with a favorable OS. Including both treatment types allowed us to give an accurate representation of the real-life setting and allowed us to investigate immune cell levels in both groups. Another limitation of our study is that we only determined *CCNE1* amplification and non-*BRCA*mut HRD profile of patients who had no *BRCA* mutation. In the literature, *CCNE1* gain/amplification and *BRCA* mutation have shown synthetic lethality [3,48]. However, we did identify 69 patients with both a *CCNE1* amplification/gain and a non-*BRCA*mut HRD profile. This is the first study reporting on the presence of *CCNE1* amplification and non-*BRCA*mut HRD simultaneously in a large group of HGSOC (20%). Our results implicate that the tumor microenvironment in this group is similar to non-*BRCA*mut HRD, however, the OS of this double classifier group is worse than OS in the non-*BRCA*mut HRD group and shows similarities to the group with *CCNE1* amplification. The exact prognostic and clinical relevance, and underlying mechanisms of this finding are yet unclear. Finally, the extent of the influence of immune cell densities within the molecular profiles could not be determined due to inadequate power.

## 5. Conclusions

Our results emphasize that HGSOC does not reflect one entity, but comprises different variants based on molecular profiles and tumor microenvironment, which in the future is ideally translated into tailored treatment approaches. Further research is warranted to clarify to what extent molecular profiles are correlated with therapy sequence and response to current targeted therapeutic modalities including PARPi. Additional research is necessary regarding treatment strategies for not only *BRCA* and non-*BRCAmut* HRD patients but also for *CCNE1* amplified patients. The present study further classified HGSOC into molecular and immunological profiles, which could serve as a basis for future research on new treatment modalities.

## Figures and Tables

**Figure 1 cancers-14-05965-f001:**
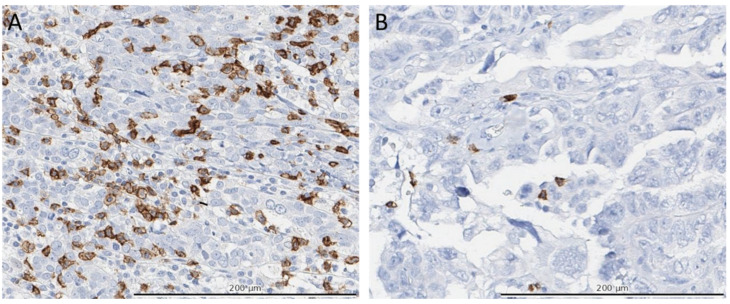
Immunohistochemical (IHC) staining of CD8+ cells. Panels (**A**,**B**) depict representative areas of the IHC stained TMA’s for CD8+ cells (brown colored cells) of two separate patients. Panel (**A**) shows a high CD8+ density (>100 cells in the entire TMA) and panel (**B**) a low density (<20 cells in the entire TMA). Total numbers of CD8+ cells were manually counted per core and scored as 0, 1–5, 6–19, 20–49, 50–100, or >100 positive cells and categorized in low, medium, and high densities based on the 25th and 75th percentiles and the median of all patients.

**Figure 2 cancers-14-05965-f002:**
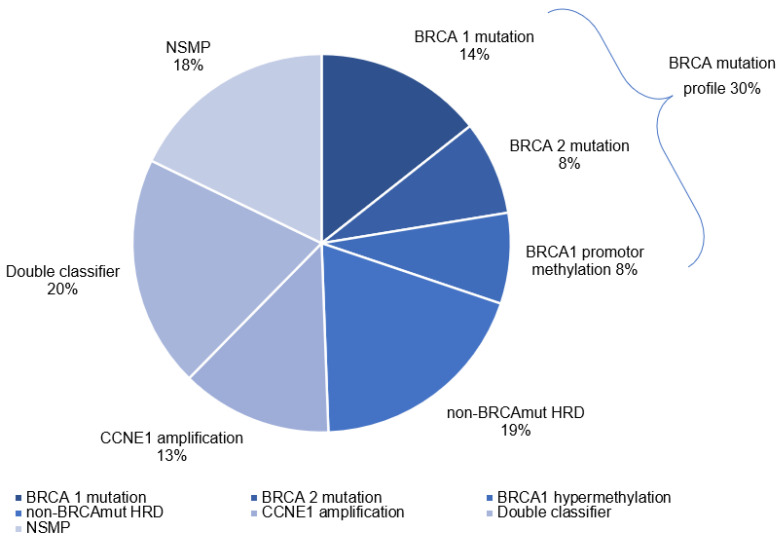
Pie chart depicting the five molecular profiles. The pie chart consists of (1) BRCA mutation or BRCA1 promotor methylation, (2) non-BRCAmut HRD, (3) CCNE1 gain/amplification, (4) double positive (non-BRCAmut HRD and CCNE1 gain/amplification), and (5) no specific molecular profile (NSMP).

**Figure 3 cancers-14-05965-f003:**
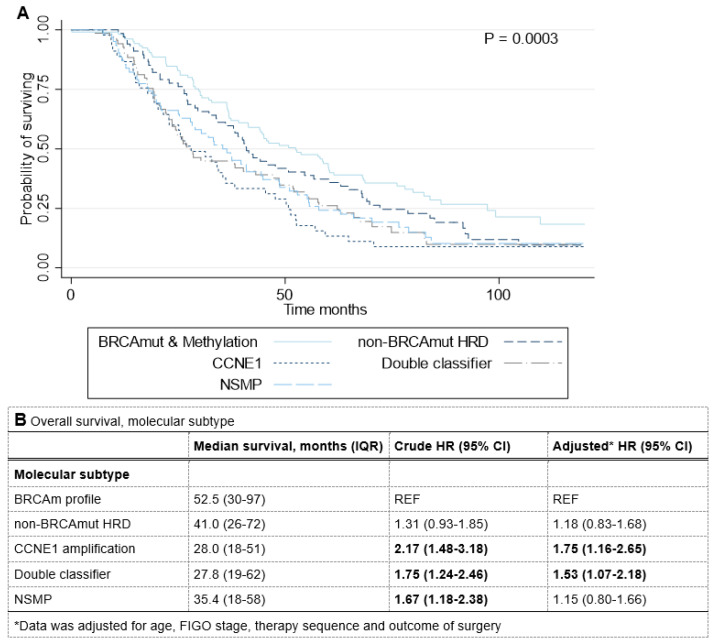
Survival analyses of patients with HGSOC, according to molecular profiles. (**A**) Kaplan–Meier curves for overall survival according to molecular profiles. *p* values were derived with the use of the log-rank statistic. (**B**) Multivariable analysis for overall survival per molecular subtype. Abbreviations: BRCAm profile: BRCA mutation or BRCA1 promotor methylation; HRD: homologous repair deficient; Double classifier: non-BRCAmut HRD and CCNE1 gain/amplification; NSMP: no specific molecular profile; CI: Confidence interval; IQR: interquartile range; HR: Hazard ratio; REF: reference; FIGO: International Federation of Gynecology and Obstetrics.

**Figure 4 cancers-14-05965-f004:**
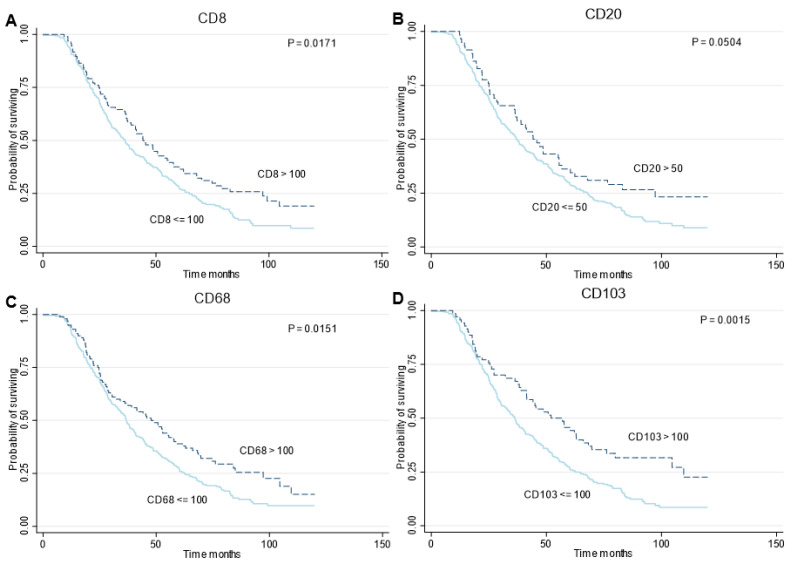
Survival analyses of patients with HGSOC, according to immune cell densities. Panels A to D show Kaplan–Meier curves for overall survival according to density of CD8 cells (**A**), CD20 (**B**), CD68 (**C**) and CD103 (**D**), respectively. *p* values were derived with the use of the log-rank statistic.

**Table 1 cancers-14-05965-t001:** Patient characteristics.

	BRCAm Profile	Non-BRCAmut HRD	CCNE1 Amplification	Double Classifier	NSMP	*p*-Value
	N = 105 (30.2%)	N = 67 (19.3%)	N = 45 (12.9%)	N = 69 (19.8%)	N = 62 (17.8%)	
**Age**						**<0.001**
<65	69 (65.7)	29 (43.3)	16 (35.6)	23 (33.3)	21 (33.9)	
65–75	31 (29.5)	30 (44.8)	14 (31.1)	33 (47.8)	22 (35.5)	
>75	5 (4.8)	8 (11.9)	15 (33.3)	13 (18.8)	19 (30.7)	
Median (IQR)	60 (53–66)	66 (61–71)	71 (62–77)	68 (63–73)	68 (61–76)	**<0.001**
**FIGO stage**						
II	7 (6.7)	5 (7.5)	1 (2.2)	5 (7.3)	0 (0.0)	0.598
III	68 (64.8)	41 (61.2)	31 (68.9)	41 (59.4)	37 (59.7)	
IV	29 (27.6)	19 (28.4)	11 (24.4)	21 (30.4)	24 (38.7)	
Unknown	1 (1.0)	2 (3.0)	2 (4.4)	2 (2.9)	1 (1.6)	
**Treatment sequence**						
PDS	49 (46.7)	27 (40.3)	19 (42.2)	32 (46.4)	13 (21.0)	0.014
NACT-IDS	56 (53.3)	40 (59.7)	26 (57.8)	37 (53.6)	49 (79.0)	
**Surgical outcome**						
Complete	73 (69.5)	37 (55.2)	21 (46.7)	34 (49.3)	29 (46.8)	0.141
Optimal	22 (21.0)	25 (37.3)	19 (42.2)	26 (37.7)	24 (38.7)	
Suboptimal	6 (5.7)	3 (4.5)	5 (11.1)	7 (10.1)	8 (12.9)	
Unknown	4 (3.8)	2 (3.0)	0 (0.0)	2 (2.9)	1 (1.6)	

Abbreviations: FIGO: International Federation of Gynecology and Obstetrics; BRCAm profile: BRCA mutation or BRCA1 promotor methylation; HRD: homologous repair deficient; Double classifier: non-BRCAmut HRD and CCNE1 gain/amplification; NSMP: no specific molecular profile; PDS: primary debulking surgery; NACT-IDS: neoadjuvant chemotherapy and interval debulking surgery.

**Table 2 cancers-14-05965-t002:** Immune cell composition stratified by molecular profiles.

	BRCAm Profile	Non-BRCAmut HRD	CCNE1 Amplification	Double Classifier	NSMP	*p*-Value
	N = 105 (30.2%)	N = 67 (19.3%)	N = 45 (12.9%)	N = 69 (19.8%)	N = 62 (17.8%)	
**CD8+ cells**						0.082
<20	13 (12.4)	13 (19.4)	13 (28.9)	14 (20.3)	14 (22.6)	
20–100	58 (55.2)	42 (62.7)	23 (51.1)	39 (56.5)	26 (41.9)	
>100	34 (32.4)	12 (17.9)	9 (20.0)	16 (23.2)	22 (35.5)	
**CD103+ cells**						
<20	28 (26.7)	26 (38.8)	19 (42.2)	21 (30.4)	19 (30.7)	0.290
20–100	48 (45.7)	28 (41.8)	21 (46.7)	36 (52.2)	32 (51.6)	
>100	29 (27.6)	13 (19.4)	5 (11.1)	12 (17.4)	11 (17.7)	
**CD20+ cells**						
<20	55 (52.4)	38 (56.7)	31 (68.9)	40 (58.0)	31 (50.0)	0.232
20–50	28 (26.7)	22 (32.8)	12 (26.7)	18 (26.1)	18 (29.0)	
>50	22 (20.9)	7 (10.5)	2 (4.4)	11 (15.9)	13 (21.0)	
**CD68+ cells**						
<50	42 (40)	41 (61.2)	24 (53.3)	44 (63.8)	40 (64.5)	<0.001
50–100	16 (15.2)	11 (16.4)	12 (26.7)	9 (13.0)	12 (19.4)	
>100	47 (44.8)	15 (22.4)	9 (20)	16 (23.2)	10 (16.1)	

Abbreviations: BRCAm profile: BRCA mutation or BRCA1 promotor methylation; HRD: homologous repair deficient; Double classifier: non-BRCAmut HRD and CCNE1 gain/amplification; NSMP: no specific molecular profile.

**Table 3 cancers-14-05965-t003:** Progression-free survival analysis, molecular subtype.

	Median Progression Free Survival, Months (IQR)	Crude HR (95% CI)	Adjusted * HR (95% CI)
Molecular Subtype			
BRCAm profile	22.3 (15–60)	REF	REF
non-BRCAmut HRD	16.9 (11–47)	1.28 (0.89–1.86)	1.22 (0.84–1.79)
CCNE1 amplification	14.8 (12–21)	**1.94 (1.29–2.91)**	**1.57 (1.02–2.43)**
Double classifier	14.7 (11–29)	**1.90 (1.33–2.72)**	**1.77 (1.22–2.56)**
NSMP	16.7 (11–32)	**1.60 (1.12–2.30)**	1.05 (0.72–1.55)

* Data was adjusted for age, FIGO stage, therapy sequence and outcome of surgery. Abbreviations: BRCAm profile: BRCA mutation or BRCA1 promotor methylation; HRD: homologous repair deficient; Double classifier: non-BRCAmut HRD and CCNE1 gain/amplification; NSMP: no specific molecular profile; CI: Confidence interval; IQR: interquartile range; HR: Hazard ratio; REF: reference.

**Table 4 cancers-14-05965-t004:** Overall survival analysis, immune cell composition.

	Median Survival, Months (IQR)	Crude HR (95% CI)	Adjusted * HR (95% CI)
**CD8**			
≤100	36.3 (21–65)	REF	REF
>100	44.2 (25–97)	**0.73 (0.56–0.95)**	**0.72 (0.55–0.94)**
**CD20**			
≤50	36.9 (21–68)	REF	REF
>50	44.2 (25–97)	0.72 (0.52–1.00)	**0.71 (0.51–0.98)**
**CD68**			
≤100	36.8 (21–63)	REF	REF
>100	47.7 (24–97)	**0.72 (0.56–0.94)**	0.94 (0.71–1.23)
**CD103**			
≤100	36.3 (22–63)	REF	REF
>100	52.1 (25–110)	**0.61 (0.45–0.83)**	**0.68 (0.50–0.93)**

* Data was adjusted for age, FIGO stage, therapy sequence and outcome of surgery. Abbreviations: CI: Confidence interval; IQR: interquartile range; HR: Hazard ratio; REF: reference.

## Data Availability

Data sharing of anonymous clinical data from the NCR will be considered for non-commercial, research, or statistical-based use on a case-by-case basis (to be requested and approved by the NCR; gegevensaanvraag@iknl.nl). The human sequence data and tumor tissue data generated in this study are not publicly available due to patient privacy requirements but are available upon reasonable request from the corresponding author.

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
