# Peer review of "Homologous Recombination Deficiency and Cyclin E1 Amplification Are Correlated with Immune Cell Infiltration and Survival in High-Grade Serous Ovarian Cancer"

_cancers, 2022, doi:10.3390/cancers14235965_

Round 1
Reviewer 1 Report
This study shows a very interesting combination of mutation analysis of ovarian cancer with the detection of immune cells infiltrating the tumour site in a large cohort of patients. This combination enables the diversification of groups of patients according to their prognosis. The paper is well-prepared and supported by experimental data with correct statistical analysis. It can be very helpful for the refinement of diagnostics including the therapeutic consequences. Figs 3. and 4 should be submitted in a better resolution because they are not sharp. This paper is very important from a practical point of view and suitable for publication after the mentioned figures improvements.
Author Response
We kindly thank the reviewer for reviewing our article. We agree with the reviewers concern regarding the resolution of the figures and have updated the resolution. The figures were also uploaded in the requested ZIP file at high resolution and will be uploaded again (in which the resolution can be seen better).
Reviewer 2 Report
Homologous recombination deficiency and Cyclin E1 amplification are correlated with immune cell infiltration and survival in high-grade serous ovarian cancer
Summary: In this paper, the authors have presented a novel work on the genetic makeup of high-grade serous ovarian carcinoma (HGSOC) and its relationship with overall survival of patients and explored mechanism on immune cell infiltration. The authors use 360 patients with HGSOC stage IIb-IV previously treated with anticancer therapy or surgery and conducted analysis on tissue with microarrays, immunohistochemical staining of immune cells, molecular analysis of BRCA1 promoter, whole genome sequencing and classifying genetic profiles of non-BRCA mutations and CCNE1 groups to demonstrate their findings.
They have reported that the frequency of different molecular profiles of the tumor, immune cell densities stratified by molecular profiles with BRCA mutation being the most common. The BRCAm profile showed highest amount of CD8+ immune cell infiltration and CCNE1 amplification category showed the lowest and confirm that increased immune cell infiltration increases overall survival (OS). Their study reports and improved OS in patients with BRCA mutation, BRCA gene promoter methylation and non-BRCA mutation with homologous recombination deficiency and worse OS in CCNE1 amplification/gain. The authors have presented their work in an excellent manner and answered major queries in the limitations section and formed basis for more work in the field.
Minor comments:
1. Line 144, please remove additional full stop
2. Adding the abbreviation for PFS (progression free survival) be added again in line 375 and 413 would be helpful
3. The text in the figures 1,2, 3, 4 and the labelling A, B, C, D could be edited to have consistency of same font and size
4. Could the authors speculate why the ‘presumably low numbers’ of CD20+ in Figure 4B (line 410)
5. FIGO abbreviation could be explained in the text and tables
6. Figure 2, 3 and 4 could be re-checked if they are 300DPI as they appear blurry compared to Supplementary Figure S2, if possible, the color scheme and font could be kept constant.
7. Figure 4 could have a title of the immune cells used on the KM curve, just like in Fig S2.
8. Line 471 could be explained well, what is BRCA wild type? Did the BRCA wild type ovarian cancer patients have CCNE1 mutation or other mutations.
9. The NSMP group has been used to compare the OS for BRCA non mut and BRCA, could it be briefly explained in the introduction, what comprised of the NSMP group, is it a mix of various mutations or what is the basis of naming it NSMP.
10. The authors mention in line 466-468 about DNA DS break inducing chemotherapy in breast cancer, has this been done in Ovarian cancer also, could this benefit patients of CCNE1 gain, could a knockout / gene therapy of CCNE1 be a possible therapy for these patients with lower OS.
11. Could the authors be able to briefly speculate the future work needed in the field, specifically how their results could be rapidly translated to the clinical and to make next generation prognostic marker and therapy.
